# Telecommuting, Off-Time Work, and Intrusive Leadership in Workers’ Well-Being

**DOI:** 10.3390/ijerph18073330

**Published:** 2021-03-24

**Authors:** Nicola Magnavita, Giovanni Tripepi, Carlo Chiorri

**Affiliations:** 1Postgraduate School of Occupational Medicine, Università Cattolica del Sacro Cuore, 00168 Roma, Italy; 2Department of Woman/Child & Public Health, Fondazione Policlinico Universitario Agostino Gemelli IRCCS, 00168 Roma, Italy; 3CNR-IFC Research Unit of Reggio Calabria, Institute of Clinical Physiology IFC, Italian National Research Council CNR, R Via Vallone Petrara, 65, 89124 Reggio Calabria, Italy; gtripepi@ifc.cnr.it; 4Department of Educational Sciences, University of Genova (Italy), 16126 Genova, Italy; carlo.chiorri@unige.it

**Keywords:** smart work, psychosocial stressors, health promotion, work-related stress, COVID-19, anxiety, depression, happiness

## Abstract

Telecommuting is a flexible form of work that has progressively spread over the last 40 years and which has been strongly encouraged by the measures to limit the COVID-19 pandemic. There is still limited evidence on the effects it has on workers’ health. In this survey we invited 905 workers of companies that made a limited use of telecommuting to fill out a questionnaire to evaluate intrusive leadership of managers (IL), the request for work outside traditional hours (OFF-TAJD), workaholism (Bergen Work Addiction Scale (BWAS)), effort/reward imbalance (ERI), happiness, and common mental issues (CMIs), anxiety and depression, assessed by the Goldberg scale (GADS). The interaction between these variables has been studied by structural equation modeling (SEM). Intrusive leadership and working after hours were significantly associated with occupational stress. Workaholism is a relevant moderator of this interaction: intrusive leadership significantly increased the stress of workaholic workers. Intrusive leadership and overtime work were associated with reduced happiness, anxiety, and depression. These results indicate the need to guarantee the right to disconnect to limit the effect of the OFF-TAJD. In addition to this, companies should implement policies to prevent intrusive leadership and workaholism.

## 1. Introduction

The COVID-19 pandemic, caused by the SARS-CoV-2 virus, has prompted governments of several countries worldwide to encourage a type of work known as “telecommuting”. Telecommuting is also commonly referred to as “telework”, “flexible work”, “remote work”, “distance work”, “distributed work”, “virtual work”, and “flexplace” [1]. In Italy, telecommuting is often called “agile” or “smart working”, terms that actually refer to a much broader landscape than telecommuting. This synecdoche is used to mean this new way of organizing work, and it is commonly adopted by Italian companies and mass media.

Telecommuting began in the 1970s in order to alleviate traffic problems and reduce energy consumption [2], but it spread more quickly when advances in technology enabled mobile connections at increasingly affordable rates. The fact that, in developed countries, more than a third of jobs can now be performed entirely at home [3] has further promoted telecommuting. Besides the well-known advantages of this type of work (e.g., the possibility of employing disabled workers), telecommuting currently has the added function of limiting the spread of the COVID-19 pandemic while simultaneously maintaining an acceptable level of economic activity. Mathematical models have shown that changing the work style of 55% of the working population to telecommuting would be an effective way of controlling the COVID-19 pandemic [4]. Consequently, most countries are encouraging the extensive use of remote working during the current pandemic. 

Nevertheless, there is considerable debate among the scientific community regarding the advantages and drawbacks of this new form of work organization. On the one hand, in a meta-analysis of 46 studies, Gajendran and Harrison [1] concluded that telecommuting has beneficial effects on perceived autonomy, job satisfaction, performance, turnover intent, and the role stress of employees. Moreover, it may have a positive effect on work/family conflict. Telecommuting can also lead to beneficial effects on physical health: studies showed that non-telecommuters are at greater risk of obesity, alcohol abuse, physical inactivity, and tobacco use than telecommuting workers [5]. On the other hand, some researchers are concerned about the mental health of telecommuters and argue that stressors associated with telecommuting can eventually lead to exhaustion and burnout [6]. Ergonomic problems may arise in home working, although in many cases companies continue their normal strategies for workplace health and safety without considering the ergonomic situations of domestic workplaces [7]. Clearly, the occupational health and safety risks of this new work environment need to be fully assessed and understood [8,9]. 

To evaluate the effects that telecommuting has on occupational well-being and health, some organizational aspects of this type of work need to be taken into account. First, the changes it has brought about in the work timetable should be considered. The reorganization of work often requires employees to be active not only in regular working time, but also during off-time (in the evening/at night or the weekend and during holidays). Arrangements that entail non-standard working hours can potentially encroach on highly important family, social, and leisure time [10]. Data reported by European Surveys on working conditions indicated that additional off-time working from home may interfere with biological rhythms (e.g., sleep–wake cycle) and social interactions [11]. The possible appearance of family issues must also be taken into consideration, although this does not necessarily lead to demonstrable health problems [12]. 

Another concern is the isolation of teleworkers. In cases where telecommuting covers most working hours, occupational isolation can reduce organizational and social support, as well as participation in decision-making and task autonomy. This in turn may lead to an increase in occupational stress and to a reduction of job satisfaction [13,14,15]. However, these problems do not usually arise when telecommuting occupies only a part of the working timetable, and they can be prevented by carefully programming telecommuting and by organizing frequent meetings between managers and workers. The style of leadership is also of the utmost importance in guiding the organizational evolution from traditional to flexible work and can influence the final health outcomes. As reported by Contreras et al. [16] some leaders and supervisors may tend to adopt an ”intrusive” leadership style, namely, the exploitation of employees with work and information overload that overlap with domestic and work settings, resulting in an intrusion into employees’ personal life that, in turn, can yield negative mental health outcomes.

In addition to organizational factors (such as leadership and demand for off-time work) that can influence the effect telecommuting has on the well-being and mental health of workers, there are also individual aspects that must be taken into consideration. A reorganization of work and the availability of time saved by not commuting can induce many workers to increase their work commitment [17]. This may result either in workaholism (WA), i.e., a pathological addiction to work, or work engagement, i.e., a healthy positive pattern of thoughts and feelings about one’s job [18]. These different approaches have different effects on perceived interference between work and life domains [19] and on workers’ well-being. 

The complexity of this framework explains why, after many decades of studies on telecommuting, there is little evidence regarding the effects that this reorganization of work has on the health and well-being of workers. New studies are needed to help bridge the knowledge gap. A distinction should be made between studies focusing on situations where telecommuting occupies a limited part of working hours and those that investigate situations in which telecommuting is prevalent or the only type of work, as it has occurred during recent pandemic lockdowns. In the former it is possible to evaluate the effect of organizational and individual factors without taking into account the effect of social isolation that occurs in the latter.

Long before lockdown measures were adopted and telecommuting became the only possible form of productivity during the current pandemic, many companies had introduced forms of telecommuting that usually comprised only a small part of the overall working time. In this study our aim was to evaluate how workers employed in companies that made limited use of telecommuting were affected by occupational stress, happiness, and common mental issues (CMIs), i.e., relatively high levels of anxiety and depression symptoms. 

In this study we used structural equation modeling (SEM) to test two hypotheses. First, we hypothesized that an intrusive leadership style that did not respect workers’ privacy and that demanded off-time work was associated with occupational stress, which, in turn, was related to reduced happiness and CMIs. Second, we investigated whether a workaholic attitude could increase the negative effects of intrusive leadership and demand for off-time work. 

The research was performed in 2019, before the effects of the pandemic extended the application of telecommuting, carrying out a survey in companies that had used telecommuting and off-time working only for organizational reasons and not to counter the spread of COVID-19 infection.

## 2. Materials and Methods

### 2.1. Participants

We selected 17 trade and service sector companies in which telecommuting activities accounted for a limited part of the working time (<10 h per week) to take part in our study. We chose companies telecommuting for a modest share of their employees’ weekly hours because, based on a meta-analysis of 28 primary studies, Gajendran and Harrison [1] reported that the relationship between the extent of telecommuting and job satisfaction is curvilinear, such that satisfaction and amount of telecommuting are positively related at lower levels of telecommuting, but satisfaction plateaus at higher levels of telecommuting (around 15.1 h per week). High-intensity telecommuting may cause isolation, thus damaging relationships with coworkers [1]. We wanted to exclude the effect of isolation, and only study the effect of leadership and overtime work, which may be present in all types of telecommuting, regardless of duration. The companies were situated in the Italian region of Latium. During their routine fitness for work medical examination, 910 workers from these companies were asked to fill out a questionnaire. Nine hundred five (99.4%) accepted. Of these, 331 workers (36.6%) were males; the mean age was 45.93 (*SD* = 11.39, range 20–72) years. The study was conducted in compliance with the Helsinki Declaration (as revised in Brazil, 2013) and was authorized by the University Ethical Committee (ID 3008). Prior to participation, all participants gave their written informed consent.

### 2.2. Measures

The frequency of off-work hours technology-assisted job demand (OFF-TAJD) [20] was measured using three questions (e.g., How often does your organization require you to answer the phone and emails during the holidays?), using a 5-point Likert-type scale with responses graded from 1 = “never” to 5 = “always”, which were used as indicators for an off-time work (OW) latent factor in the SEM model (omega reliability 0.91 [0.89, 0.92]).

Intrusive leadership style was assessed using three questions derived from the Toxic Leadership Scale developed by Schmidt [21] (e.g., “Does your supervisor invade the privacy of employees?”). Participants were asked to report the frequency with which they experienced intrusive leadership by their supervisors using a 5-point Likert-type scale ranging from 1 = “never” to 5 = “always”. The three items were used as indicators for an intrusive leadership (IL) latent factor in the SEM model (omega reliability 0.80 [0.78, 0.83]).

Workaholism (WA) was assessed using the Bergen Work Addiction Scale (BWAS) [22]. The BWAS comprises seven items that tap into the main components of work addiction, such as the salience of work (i.e., it dominates thinking and behavior); the effect of work in modifying mood; the tolerance developed by the individual (i.e., whether increasing amounts of the work are needed to achieve initial effects); the presence of withdrawal experiences (e.g., unpleasant feelings when the activity is discontinued); experiences of conflict in social relationships and other activities; relapse experiences (i.e., tendency for reversion to earlier patterns of the activity after abstinence or control); and some kind of health and/or other problem for the addicted worker. Each item is rated on a 5-point Likert-type scale ranging from 1 = “never” to 5 = “always”. Following Little et al. [23], we combined the items into three parcels, i.e., aggregate-level indicators comprising the sum (or average) of two or more items. Although this practice is advised against when the focus of the analysis is to investigate the exact relations among the individual items comprising the measured variables (e.g., when testing the psychometric properties of a scale), it is instead recommended when the focus is principally on the relations among latent variables, and item indicators are merely tools that allow one to build a measurement model for a latent construct [23], as it was the case for the SEM described in the manuscript. We thus constructed three parcels as indicators of a WA latent factor using random assignment of BWAS items (parcel 1: items 1 and 7; parcel 2: items 4 and 5; parcel 3: items 2, 3, and 6). Omega reliability was 0.81 [0.79, 0.83].

Perceived occupational stress was measured according to the Siegrist Effort/Reward Imbalance (ERI) model [24], using the short questionnaire (10 questions, with answers on a 4-point Likert-type scale) which provides scores on the effort (3 questions) and reward (7 questions) subscales [25,26]. Perceived stress was calculated as the weighted ratio between effort and reward mean scores, with higher scores indicating higher levels of stress. This score was used as an observed outcome in the SEM model.

Common mental issues (CMIs) were assessed by the Goldberg Anxiety and Depression Scale (GADS) [27,28] by means of a questionnaire consisting of 9 + 9 binary questions (e.g., “Have you felt very worried?” for anxiety; “Have you lost confidence in yourself?” for depression) that enabled us to make an assessment of the level of anxiety and depression symptoms. Three parcels per scale were computed to be used as indicators of anxiety (ANX) and depression (DEP) latent variables in the SEM model. Anxiety parcels comprised items 4, 5, and 6 (parcel 1); 1, 3, and 8 (parcel 2); and 2, 7, and 9 (parcel 3), with an omega reliability of 0.86 [0.84, 0.87]. Depression parcels comprised items 4, 6, and 8 (parcel 1); 1, 3, and 7 (parcel 2); and 2, 5, and 9 (parcel 3), with an omega reliability of 0.82 [0.80, 0.84].

Happiness (HAP) was measured with a single-item (“Do you feel happy in general?”) ranging in score from 0 to 10, according to the method indicated by Abdel-Khalek [29]. This variable was specified in the SEM model as an observed outcome.

### 2.3. Statistics

The analyses were carried out using Mplus [30]. The SEM model specified to test the hypotheses of this study is shown in Figure 1. First, we specified the association of each latent factor with its indicators (measurement models) as described in Section 2.2. Then, we specified the structural coefficients, i.e., the direct effects of the predictors (OW, WA, sex, and age) on the outcomes (anxiety, depression, happiness, ERI) to test the association between outcomes, which allowed us to test the first hypothesis. Finally, we specified the interaction effects of OW and WA on the one hand, and WA and IL on the other, to test the second hypothesis, namely, the moderating effect of WA. For the latter purpose we used the XWITH command in Mplus, which allows the direct specification of interaction effects of latent variables, instead of resorting to the specification of latent variables using item products. The method we used avoids the possible biases due to the specific item products used, but requires numerical integration, which, in turn, prevents the computation of common fit indices for the model. However, as the investigation of interaction effects of latent variables was the focus of this work, we chose the former solution to obtain their best estimates. Full information maximum likelihood estimation with robust standard errors account for missing data and relatively non-normality of the data. Latent variables were scaled by fixing the loading of the first item to 1.0 and estimating the latent variable variance, per Mplus 7 defaults.

## 3. Results

The measurement models for the latent variables showed significant (*p* < 0.001) estimates for all the parameters, suggesting that latent variable scores were adequately determined (Table 1). The latent predictors were all significantly and positively correlated with each other (WA with OW: 0.48, *p* < 0.001; WA with IL: 0.40, *p* < 0.001; IL with OW: 0.33, *p* < 0.001).

The results for the structural coefficients of the SEM are reported in Table 2 and allowed us to test the two hypotheses of this study. ERI tended to increase with OW, IL, WA, and age; happiness tended to increase with OW, was higher in males, and tended to decrease with WA and age; anxiety tended to increase with IL, WA, and age, and was higher in females; depression tended to decrease with OW and to increase with WA and age, and was higher in females. The only significant interaction effect was the WA × IL effect on ERI, which was due to a stronger effect of IL on ERI in workers with higher levels of WA (Figure 2).

Table 3 shows the correlations between the outcomes, which were inspected to test the second part of the first hypothesis, i.e., whether occupational stress, beyond being predicted by OW, WA, and IL and their interaction, was related to reduced happiness and CMIs. All correlations were statistically significant and consistent with this hypothesis, as ERI, anxiety, and depression were positively correlated, and all of them were negatively associated with happiness.

## 4. Discussion

Even when telecommuting occupies only a fraction of working time, it can lead to organizational factors that are associated with reduced workers’ well-being and health problems. Our study showed that an intrusive leadership style can result in occupational stress, low happiness, and common mental issues (anxiety and depression), consistent with the first hypothesis, which was also supported by the pattern of association among these outcomes. Demand for after-hours work performance was associated with a highly significant increase in stress, but also with a weak increase in happiness and reduced depression. Moreover, the effect of intrusive leadership on workers’ stress was more severe in participants with higher levels of workaholism, thus supporting our second hypothesis.

The results obtained in this study, which was carried out just before the COVID-19 pandemic in companies that made a limited use of telecommuting, are in line with previous findings. An intrusive leadership style has been found to be associated with increased absenteeism and reduced productivity [31,32], low job satisfaction, and high stress [33]. Our results support the idea that IL is harmful. Furthermore, the correlation between IL and WA suggests that the style of the leader can encourage excessive involvement in work on the part of employees. In the presence of intrusive leadership, a workaholic attitude increases the level of occupational stress, and consequently also the risk of suffering the negative effects of a prolonged state of distress, e.g., CMIs. In fact, previous studies have already found that WA is associated with the occurrence of mental issues [34,35]. We also observed a weak correlation between WA and happiness. This seemingly paradoxical result deserves further investigation. The association might lie in the fact that, at least for some workers, having to work extra hours might mitigate work-induced guilt, which, in turn, can have beneficial effects on happiness and depression. For example, Hochwarter et al. [36] found that work-induced guilt negatively predicted life satisfaction, and this effect was stronger in workers with low resource management. Hence, for these workers, having more hours for working might help them to cope more effectively with their work-induced guilt, whence occurs (slightly) higher happiness and (slightly) lower depression.

Requesting work outside of office hours or during holidays (OFF-TAJD work) is another organizational factor that can be detrimental to workers. In nurses, who are often exposed to an involuntary extension of working hours, overtime is associated with reduced work engagement, depressive symptoms [37], reduced productivity [38], and physical health problems [39]. 

The study confirms the association of stress (ERI) with happiness, anxiety, and depression. The size of the association of these outcomes looks in line with previous studies indicating the inverse relationship between depression or anxiety and happiness [40], the effects of changes in ERI levels on mental health [41], and the comorbidity of anxiety and depression [42].

To the best of our knowledge, this is the first study that has focused specifically on the effect of off-time work in telecommuters. Our findings support the hypothesis that the request for off-time work during telecommuting is a negative factor for workers’ well-being.

Our study also has some relevant implications for business organizations. An intrusive leadership style is not the result of telecommuting. Toxic leadership exists in many workplaces, regardless of the type of work organization. The results of our study suggest that this type of leadership can be harmful to workers and productivity and should therefore be prevented in companies that intend to initiate telecommuting. 

The demand for work performance outside of working hours is another characteristic that is already present in many workplaces, regardless of telecommuting. The person responsible for this request is nearly always the leader. On 21 January 2019, the European Parliament approved a resolution containing recommendations to the Commission on the right to disconnect from work during non-work hours and holidays (2019/2181 (INL) [43]. Our study also indicates that this type of protection is not only necessary but also needs to be strengthened. A corporate policy to contrast intrusive and toxic leadership could be created to extend the European Parliament guidelines. 

Finally, the results of this study restate that workaholism, like all other forms of addiction, is a harmful phenomenon. Companies should consider carrying out health promotion interventions to identify and control cases of workaholism.

This study also provides some guidance for occupational health services in the workplace. Addiction to work could have a negative effect on the worker, increasing occupational stress, anxiety, and depression, and decreasing happiness. These workers should be identified by the health surveillance service in order to prevent these harmful effects. While workaholism is closely related to personal characteristics, there are some stressors or job demands that can eventually become enhancers. Stress prevention in the workplace, which in European countries is one of the employer’s duties [44], must identify and prevent these factors, including intrusive leadership style and the demand for overtime work.

While this study benefited from a large sample and elevated worker participation, some limitations need to be pointed out. Companies were selected using convenience sampling; therefore, the sample of participants cannot be considered representative of all telecommuting workers. Given that this study was carried out before the COVID-19 pandemic in organizations that chose this sort of working, the results cannot be generalized either to workers who might have found themselves telecommuting during the pandemic as a result of a need, rather than a choice. Another limitation of this study is the reliance on self-report measures, which are known to be biased by factors such as socially desirable responding. Future studies should therefore include objective measures of, e.g., stress, too.

Another strong point of our study is timing: we addressed this particularly important topic before the pandemic. The mandatory nature of telecommuting during lockdown and the considerable extension of telecommuting, even in situations that do not comply with normal office work standards/organization, would make it very difficult to study this phenomenon at the present time, since the effects of leadership style and off-time work are currently entangled with social isolation and psychological and environmental discomfort. Moreover, the psychosocial effects of quarantine and isolation [45] and the complex series of emotional reactions that underlie what is known as pandemic fatigue [46] undoubtedly have mental health effects that are not easy to distinguish from those caused by telecommuting. Our study has the advantage of having observed the situation when social isolation was absent and the effects on the mental health of workers could be uniquely ascribed to their job.

## 5. Conclusions

In conclusion, the transition from traditional office work to telecommuting is a profitable and unstoppable phenomenon. Telecommuting is certainly a practical way of improving production, integrating workers with disabilities, diminishing commuting and environmental pollution, and reducing the spread of infection. However, great attention must be given to ensuring that this type of remote working is accompanied by a correct style of leadership and respect for the privacy and needs of workers.

## Figures and Tables

**Figure 1 ijerph-18-03330-f001:**
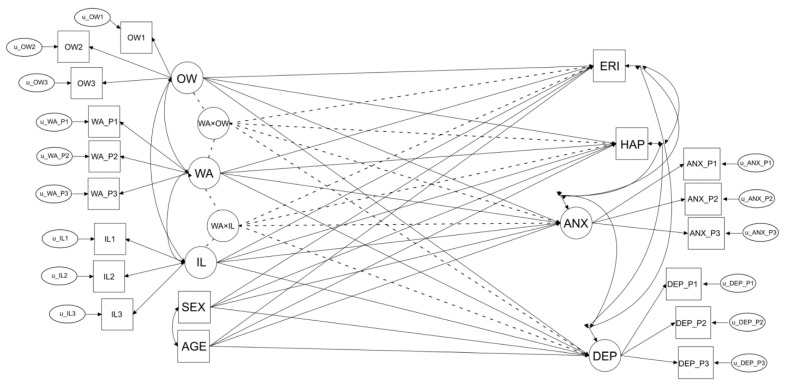
Structural equation model for the prediction of effort/reward imbalance (ERI); happiness (HAP); anxiety (ANX); and depression (DEP) using gender (SEX); age (AGE); intrusive leadership (IL); off-time work (OW); workaholism (WA); and the interactions of WA with IL (WA × IL) and OW (WA × OW). Ovals represent latent variables; rectangles represent observed variables. Solid lines represent direct effects or correlations; dotted lines represent interaction (moderation) effects. Observed variables whose name contains “_P” denote parcels of observed variables (see text for more details). Latent variables whose name contains “u_” denote uniquenesses (or residual variances) of the corresponding observed variables.

**Figure 2 ijerph-18-03330-f002:**
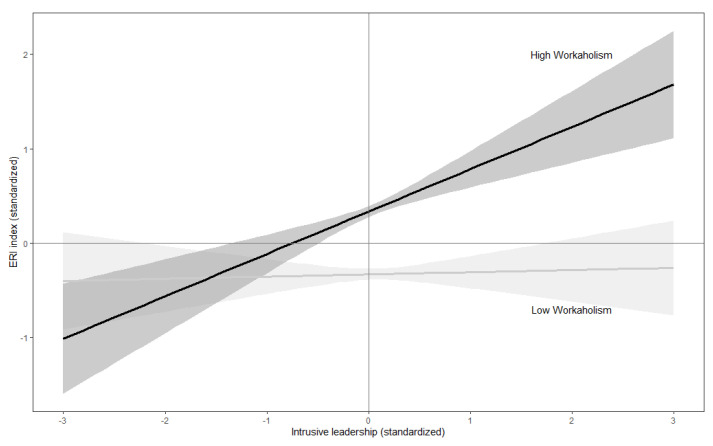
Mediating effect of workaholism on the association between intrusive leadership and effort/reward imbalance.

**Table 1 ijerph-18-03330-t001:** Parameter estimates for the measurement model of the structural equation model in Figure 1.

Observed Variable	Latent Variable	Standardized Estimate	Residual Variance
OW1	Off-time work (OW)	0.82 ***	0.33 ***
OW2	Off-time work (OW)	0.94 ***	0.12 ***
OW3	Off-time work (OW)	0.86 ***	0.26 ***
WA_P1	Workaholism (WA)	0.75 ***	0.44 ***
WA_P2	Workaholism (WA)	0.65 ***	0.58 ***
WA_P3	Workaholism (WA)	0.87 ***	0.24 ***
IL1	Intrusive Leadership (IL)	0.47 ***	0.78 ***
IL2	Intrusive Leadership (IL)	0.86 ***	0.26 ***
IL3	Intrusive Leadership (IL)	0.84 ***	0.29 ***
ANX_P1	Anxiety (ANX)	0.64 ***	0.59 ***
ANX_P2	Anxiety (ANX)	0.62 ***	0.62 ***
ANX_P3	Anxiety (ANX)	0.66 ***	0.56 ***
DEP_P1	Depression (DEP)	0.57 ***	0.68 ***
DEP_P2	Depression (DEP)	0.60 ***	0.64 ***
DEP_P3	Depression (DEP)	0.59 ***	0.65 ***

Note: ***, *p* < 0.001; Observed variables whose name contains “_P” denote parcels of observed variables (see text for more details). Residual variances correspond to elements in Figure 1 whose name contains “u_”.

**Table 2 ijerph-18-03330-t002:** Standardized parameter estimates for the structural model of the structural equation model in Figure 1.

	Outcome			
Predictor	ERI	Happiness	Anxiety	Depression
Off-time work (OW)	0.14 **	0.10 *	−0.01	−0.13 *
Intrusive Leadership (IL)	0.26 ***	−0.06	0.13 *	0.10
Workaholism (WA)	0.44 ***	−0.39 ***	0.67 ***	0.83 ***
WA × IL	0.15 **	−0.03	−0.04	0.03
WA × OW	−0.02	−0.01	0.05	0.02
Gender (Male)	0.00	0.09 *	−0.20 ***	−0.17 ***
Age	0.14 ***	−0.22 ***	0.17 ***	0.13 **

Note: ***, *p* < 0.001; **, *p* < 0.01; *, *p* < 0.05; the last two lines show the effects of the latent variable interaction.

**Table 3 ijerph-18-03330-t003:** Correlation matrix of the outcomes in the structural equation model in Figure 1.

	ERI	Happiness	Depression
Happiness	−0.16 **		
Depression	0.28 ***	−0.42 ***	
Anxiety	0.25 ***	−0.28 ***	0.88 ***

Note: ***, *p* < 0.001; **, *p* < 0.01; ERI, effort/reward imbalance.

## Data Availability

Data are deposited on Zenodo repository DOI: md5:9134e2f73a6cc182d48f9810d64bcfd6.

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
