# Peer review of "Telecommuting, Off-Time Work, and Intrusive Leadership in Workers’ Well-Being"

_ijerph, 2021, doi:10.3390/ijerph18073330_

Round 1
Reviewer 1 Report
I wish to thank the authors for the interesting article. In fact, I do not have a lot of remarks on it.
Some very minor remarks first:
page 2, line 51. I would add "On the one hand" and then continue with "in a meta-analysis". Because, generally, when you say "on the other hand", you have to mention "On the one hand" first - it's a structural language thing.
Page 3, line 118: "These effects were controlled for gender and the age of the participants". I would remove this. You shouldn't describe methods in the introduction.
Page 4, line 163: "as it was case of the SEM" - "as it was the case for the SEM"
Then, the only major remark I have, is that the analysis needs to be explained more. Describe SEM more - is this the usual type of SEM? (as I know that some SEMS use factor analysis)? Is the correlation matrix in table 3 part of the SEM you have employed? Et cetera.
In relation to that: what is the point of the correlation matrix in table 3? Is it part of the SEM? Or is it something you did yourself? If so, please include it in the method section. If not, why did you do it? The correlations are also very weak overall - despite that they significant. Significance, as you know, only refers to a chance that a result might be a coincidence. However, according to most Pearson correlation interpretations, only the correlation between depression and anxiety is strong. I also wonder in what manner these results really matter to your study? If they're important, shouldn't they be mentioned clearly in the discussion? (or did I mis that?)
Author Response
Reviewer1
I wish to thank the authors for the interesting article. In fact, I do not have a lot of remarks on it.
Some very minor remarks first:
page 2, line 51. I would add "On the one hand" and then continue with "in a meta-analysis". Because, generally, when you say "on the other hand", you have to mention "On the one hand" first - it's a structural language thing.
Page 3, line 118: "These effects were controlled for gender and the age of the participants". I would remove this. You shouldn't describe methods in the introduction.
Page 4, line 163: "as it was case of the SEM" - "as it was the case for the SEM"
Response: Thanks for these suggestions, we have corrected as suggested
Then, the only major remark I have, is that the analysis needs to be explained more. Describe SEM more - is this the usual type of SEM? (as I know that some SEMS use factor analysis)?
Response: Yes, we used a standard application of SEM, except perhaps for the specification of the latent interaction effects. We added a more detailed description of the model at the beginning of the Statistics paragraph.
Is the correlation matrix in table 3 part of the SEM you have employed? Et cetera.
In relation to that: what is the point of the correlation matrix in table 3? Is it part of the SEM? Or is it something you did yourself? If so, please include it in the method section. If not, why did you do it? The correlations are also very weak overall - despite that they significant. Significance, as you know, only refers to a chance that a result might be a coincidence. However, according to most Pearson correlation interpretations, only the correlation between depression and anxiety is strong. I also wonder in what manner these results really matter to your study? If they're important, shouldn't they be mentioned clearly in the discussion? (or did I mis that?)
Response: Table 3 shows the correlations between outcomes (in Figure 1, the double-headed arrows between ERI, anxiety, depression, and happiness), that, as now highlighted in the revision, allowed us to test second part of the first hypothesis, namely, the association of stress (ERI) with happiness, anxiety, and depression, beyond the effects of OW, IL, and WA. The size of the association of these outcomes looks in line with previous studies (for happiness, see, e.g., Spinhoven, P.; Elzinga, B. M.; Giltay, E.; Penninx, B. W. J. H. Anxious or Depressed and Still Happy? PLoS One 2015, 10 (10). https://doi.org/10.1371/journal.pone.0139912.; for ERI see, e.g., (1) Barrech, A.; Riedel, N.; Li, J.; Herr, R. M.; Mörtl, K.; Angerer, P.; Gündel, H. The Long-Term Impact of a Change in Effort–Reward Imbalance on Mental Health—Results from the Prospective MAN-GO Study. Eur. J. Public Health 2017, 27 (6), 1021–1026. https://doi.org/10.1093/eurpub/ckx068.; for the association of anxiety and depression see, e.g., Kaiser, T.; Herzog, P.; Voderholzer, U.; Brakemeier, E. Unraveling the Comorbidity of Depression and Anxiety in a Large Inpatient Sample: Network Analysis to Examine Bridge Symptoms. Depress. Anxiety 2021, 38 (3), 307–317. https://doi.org/10.1002/da.23136).

Reviewer 2 Report
Thank you for giving me the opportunity to review this article. It's an interesting paper on how telecommute some hours a week might affect the workers well-being. In the last year, the pandemic has increased the number of people who are teleworking and therefore it is a topic of great interest today. The impact of intrusive leadership on occupational stress is highlighted. They point out the mediating effect of workaholism on the association between intrusive leadership and occupational stress.
Several issues should be addressed by the authors that I briefly sum as:
In section 2.1.
Inclusion criterion was working less than 10 hours a week. Why? Why that cut-off point? It should be explained. The choice of teleworking was always voluntary? There were some exclusion criteria? It would have been interesting to know other data that might influence the results as the type of work, responsibility, position, mental or chronic illness…
In section 2.2
In OFF-TAJD paragraph: It should be included how the questions are scored
I am concerned about using just three questions to assess intrusive leadership. Could you include more information about why these questions were selected? What criteria were followed?
In section 4
The effect of intrusive leadership (IL) and work outside hours (OW) on occupational stress are clears, however work outside hours (OW) increases happiness and decreases depression (Table 2). The first paragraph of the discussion should be rewritten with these results in mind.
The authors show how addiction to work could have a negative effect on the worker, increasing occupational stress, anxiety and depression and decreasing happiness. These workers should be identified in order to prevent these harmful effects. While workaholism is closely related to personal characteristics, there are some stressors or job demands that can eventually become enhancers. I miss a more extensive discussion on this topic of great relevance to the health of workers
Author Response
Reviewer2
Thank you for giving me the opportunity to review this article. It's an interesting paper on how telecommute some hours a week might affect the workers well-being. In the last year, the pandemic has increased the number of people who are teleworking and therefore it is a topic of great interest today. The impact of intrusive leadership on occupational stress is highlighted. They point out the mediating effect of workaholism on the association between intrusive leadership and occupational stress.
Several issues should be addressed by the authors that I briefly sum as:
In section 2.1.
Inclusion criterion was working less than 10 hours a week. Why? Why that cut-off point? It should be explained. The choice of teleworking was always voluntary? There were some exclusion criteria? It would have been interesting to know other data that might influence the results as the type of work, responsibility, position, mental or chronic illness…
Response: We appreciated the comment because it allowed us to add a brief explanation of our choice in the manuscript. Based on a meta-analysis of 28 primary studies, Gajendran and Harrison (2007) reported that the relationship between the extent of telecommuting and job satisfaction is curvilinear, such that satisfaction and amount of telecommuting are positively related at lower levels of telecommuting, but satisfaction plateaus at higher levels of telecommuting (around 15.1 hours per week). high-intensity telecommuting (more than 2.5 days a week) harmed relationships with coworkers (Gajendran & Harrison 2007) The explanation for this curvilinear effect may lie in the social and professional isolation that telecommuters face when telecommuting frequently. This lack of social interaction may offset any gains in job satisfaction. In this study we were interested in evaluating the effect of two aspects that are innate with telework regardless of its duration, such as intrusive leadership and overtime work. We wanted to exclude the effect of isolation.
In section 2.2
In OFF-TAJD paragraph: It should be included how the questions are scored
Response: Thanks for this observation. We added the specification: “using a 5-points Likert scale with responses graded from never = 1, to always = 5”.
I am concerned about using just three questions to assess intrusive leadership. Could you include more information about why these questions were selected? What criteria were followed?
Response: We have considered the concept of intrusive leadership as part of the broader concept of toxic leadership, on which there are numerous contributions in the literature (Whicker 1996; Lipman-Blumen, 2005a; Wilson Starks 2003; Reed, 2004; Flynn 1999, Schmidt, 2008). By examining the questionnaire proposed by Schmidt, we isolated the three questions that were related to the concept of invasion of privacy. We limited ourselves to three questions because the questionnaires to be administered in the workplace at the time of the visits must be very short and, possibly, immediately assessable by the doctor. The three questions were as it follows: "How does your boss (if you don't have a direct boss, the person you refer to in an informal way) behave with collaborators?" 1. invades the privacy of collaborators; 2. does not respect workers' extra-work commitments; 3. tends to make urgent requests?”
In section 4
The effect of intrusive leadership (IL) and work outside hours (OW) on occupational stress are clears, however, work outside hours (OW) increases happiness and decreases depression (Table 2). The first paragraph of the discussion should be rewritten with these results in mind.
Response: we changed the first sentence, as recommended. The manuscript now is as follows: “Our study showed that an intrusive leadership style can result in occupational stress, low happiness, and common mental issues (anxiety and depression), consistent with the first hypothesis, which was also supported by the pattern of association among these outcomes. Demand for after-hours work performance was associated with a highly significant increase in stress, but also with a weak increase in happiness and reduced depression. This seemingly paradoxical result deserves further investigation. A possible explanation might lie in the fact that, at least for some workers, having to work extra-hours might mitigate work-induced guilt, which, in turn, can have beneficial effects on happiness and depression. For instance, Hochwarter et al. (2007), found that work-induced guilt negatively predicted life satisfaction, and this effect was stronger in workers with low resource management. Hence, for these workers having more hours for working might help them to cope more effectively with their work-induced guilt, whence (slightly) higher happiness and (slightly) lower depression.
The authors show how addiction to work could have a negative effect on the worker, increasing occupational stress, anxiety and depression and decreasing happiness. These workers should be identified in order to prevent these harmful effects. While workaholism is closely related to personal characteristics, there are some stressors or job demands that can eventually become enhancers. I miss a more extensive discussion on this topic of great relevance to the health of workers
Response: We are sincerely grateful for this observation which allowed us to add a paragraph on the consequences of this study for occupational medicine. The paragraph is now as follows:
“This study also provides some guidance for occupational health services in the workplace. Addiction to work could have a negative effect on the worker, increasing occupational stress, anxiety and depression and decreasing happiness. These workers should be identified by the health surveillance service in order to prevent these harmful effects. While workaholism is closely related to personal characteristics, there are some stressors or job demands that can eventually become enhancers. Stress prevention in the workplace, which in European countries is one of the employer's duties [ref] must identify and prevent these factors, including intrusive leadership style and the demand for overtime work”
